# Features of the Effect of Quercetin on Different Genotypes of Wheat under Hypoxia

**DOI:** 10.3390/ijms25084487

**Published:** 2024-04-19

**Authors:** Larisa Ivanovna Fedoreyeva, Elena Michailovna Lazareva, Neonila Vasilievna Kononenko

**Affiliations:** 1All-Russia Research Institute of Agricultural Biotechnology, Timiryazevskaya 42, 127550 Moscow, Russia; lazareva_e@yandex.ru (E.M.L.); nilava@mail.ru (N.V.K.); 2Biological Department, M.V. Lomonosov Moscow State University, Leninskie Gory 1, 119991 Moscow, Russia

**Keywords:** *Triticum aestivum* L., *Triticum durum* Desf., hypoxia, quercetin, ROS, cytological analysis

## Abstract

Hypoxia is one of the common abiotic stresses that negatively affects the development and productivity of agricultural crops. Quercetin is used to protect plants from oxidative stress when exposed to environmental stressors. O_2_ deficiency leads to impaired development and morphometric parameters in wheat varieties Orenburgskaya 22 (*Triticum aestivum* L.) and varieties Zolotaya (*Triticum durum* Desf.). Cytological analysis revealed various types of changes in the cytoplasm under conditions of hypoxia and treatment with quercetin. The most critical changes in the cytoplasm occur in the Zolotaya variety during pretreatment with quercetin followed by hypoxia, and in the Orenburgskaya 22 variety during hypoxia. Quercetin has a protective effect only on the Orenburgskaya 22 variety, and also promotes a more effective recovery after exposure to low O_2_ content. Hypoxia causes an increase in reactive oxygen species and activates the antioxidant system. It has been shown that the most active components of the antioxidant system in the Orenburgskaya 22 variety are MnSOD and Cu/ZnSOD, and in the Zolotaya variety GSH. We have shown that quercetin provides resistance only to the wheat genotype Orenburgskaya 22, as a protective agent against abiotic stress, which indicates the need for a comprehensive study of the effects of exogenous protectors before use in agriculture.

## 1. Introduction

Plants, due to their immobile nature, are very dependent on environmental conditions and suffer from its adverse effects. To counteract negative factors, plants have developed a powerful defense system. Various enzymes and numerous secondary metabolites are involved in the protective functions [1]. Wheat is one of the most important agricultural crops. The survival of half of the population of our planet depends on the yield of this crop and the quality of the grain. In mid-latitudes with a temperate climate, which is most favorable for the normal growth of wheat, there is an increase in wetland crop lands associated with heavy seasonal rainfall.

O_2_ is an essential element for plant survival. O_2_ deficiency impairs respiration and other biochemical processes [2]. Not only hypoxia, but also a number of other conditions can lead to a decrease in O_2_ availability; for example, salt stress can disrupt symplastic connections between cells.

Most terrestrial plants can temporarily survive the O_2_ deficiency that occurs during heavy rainfall, either by accelerating shoot growth or by slowing growth and entering a dormant state. Reduced O_2_ content in flooded plants causes growth retardation and cell damage. With prolonged exposure to hypoxia, these effects are enhanced [3,4]. The stage of plant development when it is immersed in water is of great importance for the effect of the damage. This is especially observed during seed germination, and early development; flooding also has a negative effect during flowering [5,6]. Plant recovery after O_2_ deficiency is also one of the most important factors determining plant tolerance to hypoxia [7]. One of the visual indicators of plant response to hypoxia is the destruction of chlorophyll, the so-called chlorosis, which occurs when there is a lack of O_2_ [8]. As a result of hypoxia, chlorophyll degradation occurs, the degree of which depends on the duration of hypoxia. However, chlorosis in plants can continue even after normal conditions are restored [9,10].

Stress factors lead to damage causing increased reactive oxygen species (ROS) production [11,12]. A significant increase in ROS is called oxidative stress. ROS generation is carried out mainly by mitochondria during oxidative phosphorylation reactions in the electron transport chain [13,14]. Changes in mitochondria lead to the release of intermembrane proteins, the disruption of the electron transport chain, changes in the transmembrane potential difference, and the formation of excess ROS [15]. An increase in lipid oxidation in mitochondrial membranes disrupts their integrity and leads to the swelling and lysis of mitochondria. This disrupts the energy supply of cells and reduces their adaptive ability to stress factors. During hypoxia, phosphatidic acid is released, which activates type D phospholipase, which triggers the regulation of the production of ROS and calcium [16]. Wheat plants tolerate hypoxia stress by regulating lipid remodeling, causing multiple changes in endogenous lipid levels [17].

The level of formation of highly toxic ROS in plant cells is controlled by antioxidants [18]. Plants have protective mechanisms against oxidative stress, leading to a decrease in the production of ROS and their destruction [19]. The antioxidant system of plants consists of enzymatic (superoxide dismutase, ascorbate peroxidase, guaiacol peroxidase, catalase, etc.) and non-enzymatic antioxidants (flavonoids, carotenoids, ascorbate, glutathione, tocopherols, etc.) [20,21].

Low-molecular-weight antioxidants belong to the secondary antioxidant system, as they are activated under severe oxidative stress, when the enzymatic antioxidant system can no longer cope with excess ROS. The main role of low-molecular-weight antioxidants is to detect and chelate free radicals. Flavonoids are one of the main classes of secondary plant metabolites and belong to the group of low-molecular-weight antioxidants [22]. More than 9000 flavonoid derivatives have been identified in various plants, which are further divided into different subfamilies depending on the modification of their basic structure [23,24]. Flavonoids contain several subgroups, including anthocyanidins, flavones, flavanols, flavanones, etc.

Quercetin (3,3′,4′, 5,7-pentahydroxyflavone) is one of the most abundant flavonoids in plants. Quercetin (Qu), classified as a phenolic compound, has strong antioxidant effects as it helps maintain oxidative balance [25]. This flavonoid is found in the outer membrane layer of the chloroplast membrane [26]. Due to this localization, Qu is involved in the regulation of the intensity of light exposure to plants [27]. Qu plays an important role in the process of protecting plants from the effects of stress, such as ultraviolet radiation or osmotic stress, which has been confirmed in numerous studies [28,29,30,31].

To increase the efficiency of agricultural production, a variety of plant protection products are widely used. However, these products often have adverse effects on the environment and human health. Therefore, more and more attention is being paid to the use of safe agents. The exogenous use of various bioregulators is very promising due to their cost-effectiveness compared to traditional breeding or transgenic approaches for increasing plant tolerance. Currently, more and more data are emerging on the use of Qu as a protective element against various stress factors [32]. However, despite the large number of publications on the effect of the exogenous application of phenolic substances, including Qu, on physiological processes occurring in plants [33], information about it in wheat seedlings is insufficient [34]. The purpose of the study was to evaluate the effectiveness of using exogenous Qu on wheat seedlings of two genotypes exposed to hypoxia. The results obtained will allow future studies to evaluate the usefulness of this flavonoid as a preventive agent that protects plants from biotic and abiotic stresses.

## 2. Results

### 2.1. Growth Condition

The deficiency of O_2_ has the greatest negative effect on seed germination and in the early stages of plant development [5,6]. The effect of hypoxia was studied on six-day-old wheat seedlings grown in a roll culture. To study the reduction of excess ROS formed during hypoxia, the treatment of seedlings of two varieties of Orenburgskaya 22 (*Triticum aestivum*) and Zolotaya (*Triticum durum*) with Qu was used before and after hypoxia for antioxidant systems, according to the scheme below (Figure 1).

In Table 1 are given the morphometric parameters of plant changes in response to O_2_ deficiency in six-day-old seedlings of two wheat varieties before and after treatment with 0.03% Qu.

The total length of seedlings of the Zolotaya variety practically does not change under hypoxia. O_2_ deficiency leads to a decrease in shoot height by 1.21 times, but, at the same time, the root lengthens by 1.17 times. Hypoxia causes dramatic morphological changes in the Orenburgskaya 22 variety. The total length of seedlings during hypoxia decreases by 1.96 times: at the same time, the length of the roots decreased, in contrast to the Zolotaya variety, by 1.15 times. The height of shoots in the Orenburgskaya 22 variety also decreased by 1.23 times, as in the Zolotaya variety. Qu does not affect the morphometric parameters of the Zolotaya variety grown under normal conditions, and also does not affect the morphometric parameters under conditions of O_2_ deficiency. However, Qu promotes the development of the Orenburgskaya 22 variety, and leads to a slight elongation of the roots by 1.05 times. Although pre-treatment with the Qu variety Orenburgskaya 22 before hypoxia (path A) is accompanied by morphometric changes, a decrease in the total length of the seedling by 1.08 times and a decrease in shoot height by 1.13 times is observed, while the length of the roots remains practically unchanged. However, treatment with Qu after hypoxia reveals the inhibition of the growth of both roots and shoots in the Orenburgskaya 22 variety. Probably, this fact indicates that, for the Orenburgskaya 22 variety, reoxygenation is the most limiting stage after hypoxia. Unlike the Orenburgskaya 22 variety, the Zolotaya wheat variety is tolerant of O_2_ deficiency; its total seedling length remains virtually unchanged and treatment with Qu does not affect the morphometric parameters.

### 2.2. Cytological Analysis

The structure of the root cells of two wheat genotypes grown under different conditions leading to the formation of vacuoles in the cytoplasm was studied. As a result, cell types with vacuoles were identified that differed in size, shape, and localization in the cytoplasm. No changes in the cytoplasm were detected in the cells of control plants of both wheat varieties (Figure 2, Table 2).

Under hypoxic conditions, with a decrease in the availability of O_2_, small and narrow, elongated vacuoles near the nuclear membrane were detected in the root cells of the Zolotaya variety. However, most cells contained giant vacuoles near the nucleus, causing the nucleus to move toward the cytoplasmic membrane. In most cells of the Orenburgskaya 22 variety, elongated vacuoles near the nucleus were detected. In some cells, irregularly shaped vacuoles were randomly located throughout the cytoplasm. Invaginations of the plasma membrane were detected on the cell surface.

When seedlings of the Orenburg variety were treated with Qu, 22 irregularly shaped vacuoles were detected only in dividing root cells. In some cells of the Zolotaya variety, large vacuoles around the nucleus are found in the cytoplasm. Most cells had one giant vacuole around the nucleus.

Pre-treatment with Qu before exposure to hypoxia on seedlings of the Orenburgskaya 22 variety is accompanied by the formation of giant vacuoles, which can lead to kernel deformation. However, there are cells with a granular structure. In the Zolotaya variety, interphase (G2) and prophase cells are characterized by the granularity of the cytoplasm at the G2 and prophase stages.

Treatment with Qu after hypoxia in cells of the Orenburgskaya 22 variety is accompanied by the formation of a fine-grained structure of the cytoplasm. In most cells, vacuoles of irregular shape are concentrated around the nucleus. In the Zolotaya variety, under the same conditions, cells with numerous small vacuoles are found, forming necklaces around the nuclei.

### 2.3. Chlorophyll Content

Plant stress negatively affects photosynthetic electron transfer in PSI and PSII and chlorophyll biosynthesis [12,35]. Therefore, the photosynthetic apparatus is a sensor of various environmental stresses, which may be responsible for the imbalance of cellular energy due to the modification of the redox status [36]. From the data in Table 3, it is clear that hypoxia leads to a decrease in the content of chlorophyll a (Chl a) and chlorophyll b (Chl b) in the leaves of the wheat variety Orenburgskaya 22 (3.3 times and 3.7 times, respectively). However, in the leaves of the Zolotaya variety, the effect of hypoxia is of a different nature; even a slight increase in the content of Chl a and b is observed. It should be noted that the content of Chl a and Chl b in the Zolotaya variety in the control is lower than in the Orenburgskaya 22 variety (1.7 times and 1.9 times, respectively). The Chl a/Chl b ratio in the Zolotaya variety is higher than in the Orenburgskaya 22 variety due to the lower Chl b content. Under O_2_ deficiency, the Chl a/Chl b ratio increased relative to the control in both the Orenburgskaya 22 wheat and the Zolotaya variety, although not as significantly.

The addition of Qu leads to an increase in the content of Chl a and Chl b in the leaves of the wheat variety Orenburgskaya 22 and has virtually no effect on their content in the Zolotaya variety. The addition of Qu before hypoxia (path A) prevents a decrease in the content of Chl a and Chl b in the leaves of the Orenburgskaya 22 variety), thereby not disrupting the process of photosynthesis, and an even higher content of Chl a and Chl b is observed than in the control. The addition of Qu after hypoxia (Figure 1, path B) leads to an increase in the content of Chl a and Chl b, but does not lead to their complete recovery. Thus, in the leaves of the Orenburgskaya 22 variety, the process of destruction of chlorophylls (chlorosis) occurs during hypoxia. Qu prevents the process of chlorosis; however, reoxygenation in the leaves is so significant that Qu does not completely prevent the degradation of chlorophylls after hypoxia.

It is interesting to note that treatment with Qu before and after hypoxia leads to a decrease in the content of Chl a and Chl b in the leaves of the wheat variety Zolotaya compared to control. It can be assumed that, in the Zolotaya variety, there is a different set of non-enzymatic antioxidants, in contrast to the Orenburgskaya 22 variety, which actively protects green pigments from destruction during hypoxia. The addition of Qu competes with other low-molecular-weight metabolites, and, as a result, an increase in the process of chlorosis is observed.

### 2.4. Content ROS

Abiotic and biotic stress factors induce a reaction in plants, during which the ROS production in cells increases and a series of cascade reactions are triggered to neutralize excess ROS. ROS targets in cells can be various biochemical and physiological reactions, as well as cellular structures, depending on the degree of the damage to which the cell metabolism is rearranged and plants acclimatize to stress conditions or one or another variant of programmed cell death (PCD) is triggered [37,38,39].

The excessive production of ROS during oxidative stress is part of many stressful situations, including hypoxia. In the transition from a normal state to hypoxia, there is an increase in the formation of ROS; plant mitochondria undergo swelling, releasing Ca^2+^ and cytochrome c. Transmission electron microscopy under hypoxic conditions in wheat roots visualized the presence of H_2_O_2_ in the apoplast and in association with the plasma membrane [40]. If, in the control, wheat mitochondria were characterized by an electron-dense matrix, intact membranes, and distinct cristae, then, under the influence of hypoxia, mitochondria are characterized by an increase in volume, a decrease in matrix density, and the disintegration of cristae associated with deep swelling [41]. Under conditions of hypoxic stress, evidence of ROS formation under conditions of low oxygen content was found [42]. The decrease in the ATP content observed during hypoxia increases the likelihood of ROS formation due to the inhibition of the mitochondrial electron transport chain [43].

Our studies have shown that cultivated wheat plants under the action of hypoxia (24 h), when staining the roots of varieties Orenburgskaya 22 and Zolotaya with the ROS marker Carboxy-H2DFF, contain different amounts of ROS production, determined by the intensity of fluorescence. In the roots of Orenburgskaya 22 wheat grown under control conditions, the fluorescence intensity is 1.22 times lower than in the roots of the Zolotaya variety (Figure 3 and Figure 4). Root zones were noted from the most intense fluorescence to its absence. The data obtained show that, during hypoxia, the most intense ROS staining is observed in the areas of the cap and fission. At the same time, compared with the control, the increase in the level of ROS is most observed in the cells of the epidermis and cortex, and, to a lesser extent, in the cells of the central cylinder (Figure 3).

O_2_ deficiency induces the formation of ROS in both wheat genotypes. But if, in the Zolotaya variety, the amount of ROS increased by only 1.06 times, then, in the Orenburgskaya 22 variety, this increase was 2.69 times. As a result of exposure to hypoxia, the ROS content in the roots of the Orenburgskaya 22 variety exceeded that in the Zolotaya variety by more than 2 times.

Qu significantly reduces the amount of ROS by 1.51 times in the roots of the Orenburgskaya 22 variety and has virtually no effect on the Zolotaya variety. Treatment with Qu, both before and after hypoxia, practically did not change the level of ROS in the roots of the Zolotaya variety. However, pre-treatment with Qu before exposure to hypoxia reduced the production of ROS in the Orenburgskaya 22 variety by 2.45 times and approached almost the level of ROS in the control roots. Although treatment with Qu decreased after hypoxia, it was not as effective as compared to pre-treatment.

### 2.5. Content H_2_O_2_

The H_2_O_2_ content in the control shoots of both wheat varieties is close in value (Figure 5). In the roots of the Zolotaya variety, the H_2_O_2_ content is slightly higher (1.1 times) than in the roots of the Orenburgskaya 22 variety. Moreover, the H_2_O_2_ content in the roots of both wheat varieties exceeds the H_2_O_2_ value in the shoots: in the Orenburgskaya 22 variety by 6.26 times, and in the Zolotaya variety—by 7.24 times. The effect of hypoxia leads to an increase in the H_2_O_2_ content in both roots and shoots. However, the accumulation of H_2_O_2_ in different organs of different wheat genotypes has a different tendency. If, in the roots of the variety Orenburgskaya 22, H_2_O_2_ increases only by 1.49 times, then, in the shoots, by 1.64, and, in the Zolotaya variety, in the roots, by 1.15 times and, in the shoots, by 1.08. Qu reduces the H_2_O_2_ content only in the roots and shoots of the Orenburgskaya 22 variety; and, in the Zolotaya variety, the added Qu even leads to an increase in H_2_O_2_ content. Treatment with Qu before exposure to O_2_ deficiency on the Orenburgskaya 22 variety resulted in a 1.28-fold decrease in the H_2_O_2_ content in roots compared to exposure to hypoxia, and, in seedlings, by 1.23-fold. Treatment with Qu after exposure to hypoxia in seedlings of the Orenburgskaya 22 variety even led to an increase in the accumulation of H_2_O_2_ compared to hypoxia by 1.13 times. Treatment of the Zolotaya variety with Qu did not lead to significant changes in the H_2_O_2_ content in the roots. It should be noted that treatment of this wheat variety with Qu was accompanied by an increase in the H_2_O_2_ content in the shoots by 1.52 times. O_2_ deficiency in the shoots of the Zolotaya variety was not accompanied by the accumulation of H_2_O_2_; however, treatment with Qu before and after hypoxia led to an increase in H_2_O_2_.

### 2.6. Antioxidant System

Various abiotic stresses lead to an overproduction of ROS in plants, which are highly reactive and toxic and cause damage to proteins, lipids, carbohydrates, and DNA, which, ultimately, leads to oxidative stress [44]. The antioxidant system protects plants from damage caused by oxidative stress [38]. The antioxidant system (AOS) includes a complex of enzymes involved in detoxification and the conversion of superoxide ions into peroxide, and then into water, and a complex of low-molecular-weight components. Therefore, the antioxidant activity (AOA) value depends on the extraction method. We used two test systems, which are based on different methods of extraction of the material under study: water and alcohol. (R)-4-[1-Hydroxy-2-(methylamino)ethyl]-benzene-1,2-diol (HMAEB) and 2,2-diphenyl-1-picrylhydrazyl (DPPH) were used as substrates (Table 4).

The AOA value in the wheat variety Orenburgskaya 22 in the roots is 1.09 times higher than in Zolotaya, and, in the shoots of the Zolotaya 22 variety, it is 1.03 times higher than in Orenburgskaya 22 (Table 3). A decrease in O_2_ levels leads to a decrease in AOA levels in both roots and shoots of both wheat genotypes. However, the decrease in AOA in the Orenburgskaya 22 variety is more significant than in the Zolotaya variety (in roots, by 1.50 and 1.14, respectively, and, in shoots, by 1.55 and 1.1, respectively).

Qu increases AOA in the roots of the Orenburgskaya 22 variety by 1.18 times, and, in the shoots, only by 1.03 times. It is interesting to note that, in the shoots of the Zolotaya variety, Qu reduces AOA by 1.03 times. Pre-treatment of the Orenburgskaya 22 variety with Qu does not protect the roots from hypoxia and AOA remains at the same level as under hypoxia. A slight increase in AOA is observed only in the shoots of this variety. And treatment of the Orenburgskaya 22 variety with Qu after O_2_ deficiency leads to an even greater drop in AOA. Similar to the Zolotaya variety, treatment with Qu after hypoxia reduces the total level of AOA in both roots and shoots. However, pre-treatment of the Zolotaya variety with Qu leads to an increase in AOA in shoots by 1.13 times.

The ARA value is significantly higher than the AOA, but its changes have the same trend as the AOA. The ARA value in the shoots of the Zolotaya variety is also higher than in the Orenburgskaya 22 variety, and lower in the roots. It should be noted that the differences in the content of the two wheat varieties in ARA values are less significant than AOA. This is probably due to the fact that the amount of low-molecular-weight antioxidants in the Zolotaya variety is higher than in the Orenburgskaya 22 variety, since the determination of ARA uses the alcoholic extraction of plant materials, which promotes the extraction of low-molecular-weight antioxidants, such as polyphenols and flavonoids.

#### 2.6.1. Content Glutathione

Glutathione (GSH) is one of the low-molecular-weight components of AOS involved in H_2_O_2_ detoxification [24]. Glutathione is a tripeptide (L-glutamyl-L-cysteinylglycine). The protective effect of GSH is accompanied by its oxidation sulfhydryl group and conversion to glutathione disulfide (GSSG). A high GSH/GSSG ratio and the activity of enzymes associated with GSH metabolism are among the markers of plant resistance to stressors [45].

The GSH content in the roots of the Zolotaya variety (2.43 mM) is 1.87 times higher than in the Orenburgskaya 22 variety (1.32 mM) (Figure 6), and, in the shoots, it is 1.42 times higher (Figure 6). A decrease in the O_2_ level leads to a sharp increase in the GSH concentration, especially in the shoots of the Zolotaya variety by 7.68 times. At the same time, in the Orenburgskaya variety, the increase in GSH content increases only by 1.21 times. Although the GSH content in the roots of the Orenburgskaya 22 variety increases more significantly than in the shoots (1.76 times), and, in the Zolotaya variety, in the roots, less significantly (2.58 times) compared to the shoots, the total GSH content in the roots of the Zolotaya variety is, nevertheless, higher by 2.75 times.

Qu has a positive effect on the Orenburgskaya 22 variety, especially on the roots: the GSH content increases by 1.77 times. At the same time, in the roots of the Zolotaya variety, Qu inhibits the synthesis of GSH: its content decreases by 1.58 times. Qu has little effect on the GSH content in the shoots of both wheat varieties. Treatment with Qu both before and after hypoxia, in the variety Orenburgskaya 22, increases the GSH content, especially in the roots (before by 1.98 times; after by 1.44 times). If treatment of the roots of the Zolotaya variety with Qu is not as effective as for the Orenburgskaya 22 variety, then this treatment has a negative effect on the shoots: the GSH content decreases (before 1.36 times; after −1.6 times).

#### 2.6.2. Expression of Genes

The antioxidant system protects plants from damage caused by oxidative stress. ROS also influence the expression of a number of genes and, therefore, control many processes such as growth, cell cycle, PCD, abiotic stress responses, pathogen defense, systemic signaling, and development. In this review, we describe the biochemistry of ROS and their sites of production, as well as the antioxidant defense mechanisms that scavenge ROS. Superoxide ion (O^2−^.) is the most toxic molecule that accumulates in plants as a result of exposure to abiotic stress. Superoxide dismutases (SODs) convert O^2-^. into H_2_O_2_ [21]. Increased SOD activity represents one of the most important protective mechanisms that reduces oxidative damage caused by stress. However, the increased H_2_O_2_ content is reactive and toxic and must be removed. The conversion of H_2_O_2_ involves a number of enzymes that convert H_2_O_2_ into H_2_O. CAT, PX, and GPX, are among the most active enzymes involved in the neutralization of H_2_O_2_ [46].

The expression of the mitochondrial *MnSOD* gene in the Orenburgskaya 22 variety is higher than in the Zolotaya variety: in the roots, by 2.42 times; in the shoots, it decreased by 1.03 times. (Figure 7). It should be noted that, in the roots of the Orenburgskaya 22 variety, *MnSOD* activity is 2.12 times higher than in the shoots. In the Zolotaya variety, there is a slight excess of *MnSOD* activity in the shoots than in the roots (1.1 times). O_2_ deficiency causes a significant increase in *MnSOD* activity in both roots and shoots of both wheat varieties. It should be noted that hypoxia stimulates *MnSOD* activity in the Zolotaya variety (2.92 times in roots; 2.2 times in shoots) more significantly than in the Orenburgskaya 22 variety (1.74 times in roots; 1.95 times in shoots).

Treatment with Qu leads to a significant increase in the level of *MnSOD* expression in the roots of the Orenburgskaya 22 variety by 2.88 times and, in the shoots, by 2.22 times. However, the changes in expression levels are less significant, especially in roots: only 1.08 times. Pre-treatment with Qu before decreasing the O_2_ content is accompanied by an increase in *MnSOD* activity in both the roots and shoots of the Orenburgskaya 22 variety (2.01 and 2.49 times, respectively). And treatment with Qu after exposure to hypoxia is not as effective as pre-treatment. In contrast to the variety Orenburgskaya 22, in the variety Zolotaya, pretreatment is much less significant; moreover, it even leads to a decrease in the level of *MnSOD* expression, especially in the roots (2.48 times).

In control samples of wheat Orenburgskaya 22, the level of expression of the chloroplast *Cu/ZnSOD* is higher than in the Zolotaya variety, especially in the roots (2.23 times). It should be noted that the *Cu/ZnSOD* activity in the shoots of the Orenburgskaya 22 variety is slightly lower than in the roots (1.09 times), and, in the Zolotaya variety, the opposite picture is observed: in the shoots, the *Cu/ZnSOD* activity is 1.35 times higher than in the roots. A decrease in O_2_ content is accompanied by an increase in the expression of *Cu/ZnSOD;* this is especially noticeable for the Zolotaya variety: in the roots, it increases by 5.02 times, and, in the shoots, by 2.64 times.

Qu stimulates the activity of *Cu/ZnSOD* especially in the roots of the Orenburgskaya 22 variety by 2.96 times. At the same time, in the shoots of the Zolotaya variety, a slight drop in *Cu/ZnSOD* activity is observed. Treatment with Qu before or after hypoxia induces different expression responses of *Cu/ZnSOD* in roots and shoots of different wheat genotypes. If, in the Orenburgskaya 22 variety, treatment with Qu leads to the activation of *Cu/ZnSOD*, especially in the shoots (pathway a, 1.92 times), then, in the Zolotaya variety, under the same conditions, *Cu/ZnSOD* activity is inhibited in the shoots.

One of the main enzymes involved in the conversion of H_2_O_2_ to H_2_O and O_2_ is catalase (CAT) [47,48]. Based on the data obtained, it follows that *CAT* is more active in shoots than in roots in both wheat genotypes: in Orenburgskaya 22 by 4.55 times; in Zolotoy by 1.45 times. It is interesting to note that *CAT* is most active in the roots of the Zolotaya variety and in the shoots of the Orenburgskaya 22 variety. Hypoxia leads to an increase in activity in the Zolotaya variety in both roots and shoots (1.59 times, and 1.47 times, respectively). If O_2_ deficiency is not accompanied by changes in the expression of *CAT* in the shoots of the Orenburgskaya 22 variety, a significant drop in *CAT* activity by 1.43 times is observed in the roots. Qu causes the activation of the *CAT* expression in the roots of Orenburgskaya 22 (1.96 times), and, in the Zolotaya variety, the inhibition of *CAT* activity is observed. In the shoots of both wheat varieties, Qu did not have a significant effect on *CAT* levels. Pretreatment with Qu leads to a sharp increase in the level of *CAT* expression in the roots of both wheat varieties, especially in the roots of the Orenburgskaya 22 variety (4.09 times). In shoots of the Zolotaya variety, there is also an increase in *CAT* activity before and after treatment with Qu, and, in shoots of the Orenburgskaya 22 variety, under the same conditions, *CAT* activity decreases.

Peroxidase (PX) is also involved in the neutralization of H_2_O_2_ [49]. The highest *PX* activity was observed in the roots of Orenburgskaya 22 compared to the roots of the Zolotaya variety (5.93 times). In shoots, there is no longer such a big difference in the *PX* activity between varieties (1.85 times). But if, in the roots and shoots of the Orenburgskaya 22 variety, the levels of *PX* expression are close, then, in the shoots of the Zolotaya variety, the level of *PX* activity is 3.07 times higher than in the roots. Hypoxia causes a sharp increase in activity in the roots of the Zolotaya variety by 5.53 times. *PX* activity in the roots of the Zolotaya variety also increases significantly with the addition of Qu (4.08 times). The level of activity during pre-treatment with Qu before reducing the O_2_ level in the roots of the Zalotaya variety also contributes to the activation of *PX*. In shoots of the Zolotaya variety, a decrease in *PX* activity is observed, especially in the case of treatment with Qu after hypoxia by 3.16 times. If O_2_ deficiency leads to a slight increase in *PX* activity in the Orenburgskaya 22 variety, then Qu, in all variants, causes a significant decrease in *PX* activity in both roots and shoots.

Another enzyme involved in the neutralization of H_2_O_2_ is glutathion peroxidase (GPX) [50,51]. *GPX* activity in control samples of wheat variety Orenburgskaya 22 in both roots and shoots is higher than in the Zolotaya variety (1.24 times and 2.20 times, respectively). It is noted that *GPX* activity is higher in the shoots of Orenburgskaya 22 than in the roots by 1.06 times, and, in the Zolotaya variety, it is higher in the roots, than in escapes at 1.66. O_2_ deficiency leads to a decrease in *GPX* activity in all variants. Qu also inhibits *GPX* activity. Only in the case of treatment with Qu before and after exposure to hypoxia in the shoots of the Zolotaya variety is an increase in *GPX* activity observed.

Glutathion-S- transferase (GST) responsible for detoxification has approximately the same level of activity in all control samples [52,53]. Changes in the growing conditions for the Orenburgskaya 22 variety practically did not lead to significant changes. A decrease in O_2_ content significantly stimulated an increase in the activity of *GST* in the Zolotaya variety: in the roots by 2.78 times; in the shoots by 1.93 times. It is interesting to note that the addition of Qu decreased the level of *GST* expression in the shoots of the Orenburgskaya 22 variety and increased it in the shoots of the Zolotaya variety.

## 3. Discussion

Oxygen (O_2_) is an important element for normal plant development [54]. O_2_ deficiency resulting from heavy precipitation impairs respiration and other biochemical processes [54]. In addition, a number of other conditions can lead to hypoxia; for example, salt stress can disrupt symplastic connections between cells, resulting in a decrease in cell permeability to O_2_ [55]

Plants are very plastic and quickly adapt to changing unfavorable environmental conditions. Depending on the mechanisms and speed of adaptation, plants can be divided into varieties tolerant and sensitive to abiotic stress. Hypoxia is one of the most important abiotic stresses. In addition, plants often experience physiological hypoxia in tissues and organs due to limited O_2_ diffusion or rapid O_2_ consumption [56].

Hypoxia-tolerant plants tolerate long-term reductions in O_2_ availability by accelerating shoot growth and thereby increasing O_2_, or, conversely, by slowing growth while conserving nutrient resources [4]. To survive O_2_ deficiency, plants use various strategies through biochemical, anatomical, and morphological changes. For example, the distribution of O_2_ from the aerial parts to the roots is facilitated by the formation of aerenchyma [57,58]. Similarly, the formation of adventitious roots can also improve O_2_ levels in plants under waterlogged conditions [59]. Meanwhile, balanced ROS production and increased antioxidant activity can improve tolerance to hypoxia and anoxia in plants [60].

Abiotic stresses lead to a decrease in the content of chlorophyll (Chl) and carotenoids, leaf necrosis, and a decrease in metabolic functions in the cell, including photosynthesis [61]. Osmotic stress associated with ion imbalance as well as hypoxia can lead to oxidative damage [12]. The analysis of chlorophyll a and b content is an excellent tool for quantifying the damage to the photosynthetic apparatus caused by abiotic stress [62].

Stress negatively affects photosynthetic electron transfer in PSI and PSII and Chl biosynthesis in plants [63]. Therefore, the photosynthetic apparatus is a sensor of various environmental stresses, which may be responsible for the imbalance of cellular energy due to the modification of the redox status.

All significant changes occurring in the roots and shoots of two wheat genotypes grown under different conditions are presented in diagrams (Figure 8, Figure 9, Figure 10 and Figure 11). Under normal development conditions, the content of Chl a and Chl b in the Orenburgskaya 22 variety is significantly higher than that in the Zolotaya variety by 1.67 and 1.92 times, respectively. It can be assumed that the slowdown in the process of photosynthesis in the Zolotaya variety is due to the slower access of O_2_ and its absorption compared to the Orenburgskaya 22 variety. From the data in the figure, it can be seen that O_2_ deficiency leads to a decrease in the content of Chl a and Chl b in the leaves of the wheat variety Orenburgskaya 22, and, in the Zolotaya variety, it even increases. Hypoxia and the associated O_2_ deficiency turned out to be not critical for the biosynthesis of Chl in the Zolotaya variety, since the Zolotaya variety is adapted to O_2_ deficiency.

When exposed to various unfavorable stresses, including hypoxia, the level of ROS in plants increases [38]. The accumulation of the fluorescent ROS marker Carboxy-H2DFFDA in the cells and tissues of the roots of wheat seedlings under the influence of oxidative stress induced by hypoxia indicates a disruption of ROS homeostasis. In the variety Orenburgskaya 22, under hypoxia, a significant increase in ROS is observed compared to the variety Zolotaya. In particular, in the Orenburgskaya 22 variety, there is a more significant accumulation of H_2_O_2_ both in the roots and in the shoots compared to the Zolotaya variety. Increased ROS production triggers antioxidant defense in wheat. In the process of evolution, plants have developed a powerful antioxidant system to protect against external influences. A decrease in the availability of O_2_ during hypoxia leads to a decrease in the overall activity of AOA, with the most significant drop in activity noted in the roots of the Orenburgskaya 22 variety, and the least significant drop occurs in the shoots of the Zolotaya variety.

AOS includes both an enzymatic complex and low-molecular-weight antioxidants. The enzyme complex mainly includes SOD, which is divided into different isoforms depending on the metal cofactor [47]. Depending on the cofactor, the localization of SOD is determined [48,49]. The most important isoform of the enzyme is MnSOD, since it is localized in mitochondria, the main sources of ROS.

Depending on the metal interacting with the active site, enzymes can be divided into three types: FeSOD, MnSOD, and Cu/ZnSOD [23]. Different isoforms of SOD with similar functions have different metal cofactors, amino acid sequences, crystal structures, and subcellular localizations, and exhibit a different sensitivity to H_2_O_2_ in vitro [64,65]. Cu/ZnSOD, which are mainly located in chloroplasts, cytoplasm, and/or extracellular space, are present in some bacteria and all eukaryotic species [66], while MnSODs are mainly found in plant mitochondria [67,68]. FeSOD are common in prokaryotes, and protozoa, usually localized in chloroplasts and plant cytoplasm [69].

The greatest activity of mitochondrial *MnSOD* is manifested in the roots of the Orenburgskaya 22 variety. O_2_ deficiency leads to an increase in the level of *MnSOD* expression; this is especially noticeable in the roots of the Zolotaya variety. However, the overall level of *MnSOD* activity remains higher in the Orenburgskaya 22 variety. Under normal growth conditions in the roots and shoots of the Orenburgskaya 22 variety, the level of activity of another SOD, the chloroplast *Cu/ZnSOD*, exceeds that of the Zolotaya variety. However, under hypoxic conditions, this form of *SOD* is most activated in the Zolotaya variety, especially in the roots. Consequently, in the Orenburgskaya 22 variety, mitochondrial *MnSOD* is most responsible for the conversion of the superoxide ion into peroxide under O_2_ deficiency, and, in the Zolotaya variety, it is the chloroplast *Cu/ZnSOD*.

Further detoxification associated with the conversion of H_2_O_2_ into H_2_O and free O_2_ occurs with the help of a number of enzymes, the most important of which is catalase [53,54]. It should be noted that catalase activity depends on the level of H_2_O_2_. However, the greatest accumulation of H_2_O_2_ occurs in the roots of the Orenburgskaya 22 variety, but, at the same time, the activity of *CAT* decreases. Although the H_2_O_2_ content in the roots of the Zolotaya variety increases slightly under hypoxia, the expression of the *CAT* gene has the greatest activity. Another enzyme involved in the detoxification of H_2_O_2_-PX also does not show an increase in activity in the Orenburgskaya 22 variety under hypoxia, in contrast to the Zolotaya variety, in which, especially in the roots, the activity of *PX* increases sharply.

Under stressful circumstances, when ROS accumulate in plants and the enzymatic AOS is unable to neutralize excess ROS, low-molecular-weight antioxidants are activated in plants. One of the most important antioxidants is GSH [21]. The main place of its localization is chloroplasts. GSH is often used as a marker of plant resistance to stressors [45]. In the roots, the GSH content of the Zolotaya variety is almost two times higher than that of the Orenburgskaya 22 variety. As a result of a decrease in O_2_ levels, an increase in GSH content is observed in all wheat genotypes, both in shoots and roots. But if the increase in GSH in the Orenburgskaya 22 variety is not so significant, then, in the roots of the Zolotaya variety, the GSH increase occurs more than seven times. Such a high GSH content in the Zolotaya variety may be due to the fact that the functioning of chloroplasts plays a special role in the Zolotaya variety under O_2_ deficiency. This assumption is also confirmed by the special activity of the chloroplast *Cu/ZnSOD* under hypoxic conditions. One of the evidence of the special role of GSH in the Zolotaya variety under hypoxia is the activation of GST-glutathione-S-transferase, an enzyme involved in detoxification together with GSH [58,59].

Using cytological analysis, different types of changes in the structure of the cytoplasm in root cells of two wheat genotypes were identified. In most Zolotaya wheat cells, under hypoxia, giant vacuoles formed around the nucleus. In cells of the Orenburgskaya 22 variety, cells with irregularly shaped vacuoles, randomly located throughout the cytoplasm, were observed. Different changes in the structure of the cytoplasm indicate different methods of protection against O_2_ deficiency. The formation of giant vacuoles probably contributes to a better adaptation to hypoxia in the Zolotaya variety compared to the Orenburgskaya 22 variety.

The greatest changes occur in the root cells of the Zolotaya variety during pretreatment with Qu (path A). These changes are associated with the formation of a fine-grained structure of the cytoplasm. Individual cells with a similar structure are also found in the variety Orenburgskaya 22. There are significantly more such cells in Orenburgskaya 22 when seedlings are treated with Qu after hypoxia.

In stressful factors, phytoprotectors are used that protect plants and mitigate the effects of stress [70]. These metabolites include phenolic compounds, one of which is Qu. Qu is a flavonoid that plays an important role in maintaining a balanced concentration of ROS in cells and in enhancing physiological functions, providing tolerance to biotic and abiotic stresses. Qu reduces the level of H_2_O_2_, and absorbs ROS, inhibiting their aggregation. The antioxidant effect of Qu was confirmed in studies by [71], in which various plant species were subjected to paraquat stress, where Qu was shown to be an effective protective agent against the harmful effects of ROS on plants. Therefore, this flavonoid has a beneficial effect on plant resistance to oxidative stress due to the inactivation of reactive oxygen and the interaction of ROS with the electron transport of chloroplasts and mitochondria [33].

The addition of exogenous Qu to the Orenburgskaya 22 variety had a beneficial effect on both morphometric parameters and Chls content (Figure 9). Moreover, a decrease in ROS and H_2_O_2_ levels was noted. The total AOA increased, and *MnSOD* and *Cu/ZnSOD* activities also increased, especially in roots. But, at the same time, the inhibition of the activity of enzymes involved in the neutralization of H_2_O_2_ was observed. One can guess that exogenous Qu activates SOD enzymes itself and actively participates in the neutralization of H_2_O_2_, inhibiting enzyme activity, participating in this process.

Unlike the Orenburgskaya 22 variety, treatment with Qu on the Zolotaya variety does not have a noticeable effect on morphometric parameters and Chls content. Moreover, a decrease in AOA and an increase in H_2_O_2_ content were noted. In addition, the content of the important AOS component for the Zolotoy variety, GSH, decreased slightly in the roots, while the activity of the *MnSOD* and *Cu/ZnSOD* genes remained virtually unchanged or even decreased. It should be noted that, in the roots of the Zolotaya variety, treatment with Qu significantly increased the activity of *PX*. It can be assumed that, in the Zolotaya variety, there is competition between endogenous GSH and exogenous Qu. As a result of this, low-molecular antioxidants were excluded from the process of H_2_O_2_ neutralization, which led to a slight increase in the H_2_O_2_ content.

Pre-treatment with Qu before the negative impact of hypoxia on the Orenburgskaya 22 variety has a positive effect (Figure 10). Although, in comparison with the sample of wheat treated with Qu, there is a decrease in morphometric parameters with further hypoxia, but, in comparison with the sample after exposure to hypoxia (without pre-treatment with Qu), the shoot height and root length of the Orenburgskaya 22 variety are higher. The chlorophyll content in shoots of Orenburgskaya 22 does not decrease with O_2_ deficiency, which also indicates the protective mechanism of Qu. It should be noted that pre-treatment with Qu helps to reduce ROS compared to hypoxia; however, compared to the control, H_2_O_2_ accumulates in the shoots of the Orenburgskaya 22 variety. The total AOA decreases, especially in the roots, and pretreatment with Qu does not improve the situation. Under these conditions, in the roots of the Orenburgskaya 22 variety, the activities of *MnSOD* and *Cu/ZnSOD*, as well as enzymes involved in the neutralization of H_2_O_2_, except for *CAT*, are significantly reduced. However, it should be noted that the GSH content simultaneously increases. It is interesting to note that a different picture is observed in the shoots: the activity of *MnSOD* and *Cu/ZnSOD* and the activity of *PX* and *GPX* increase. Thus, it can be assumed that pre-treatment with Qu on the Orenburgskaya 22 variety has a positive protective effect mainly on the development of shoots and changes the defense mechanism in the roots, placing the main emphasis on GSH.

Pre-treatment with Qu before hypoxia in the Zolotaya variety has its own characteristics. This treatment does not lead to the accumulation of ROS; the H_2_O_2_ content also does not increase in the roots, only in the shoots, while the chlorophyll content decreases. Under these conditions, in the Zolotaya variety, as well as in the Orenburgskaya 22 variety, the activity of *MnSOD* and *Cu/ZnSOD*, as well as enzymes involved in the neutralization of H_2_O_2_, with the exception of *CAT*, is significantly reduced. At the same time, the GSH content increases significantly. Thus, we can conclude that Qu does not have a positive effect on the Zolotaya variety, and hypoxia even helps to restore the normal development of the Zolotaya variety.

The process of plant reoxygenation—recovery from hypoxia—is also a stress state in plants [7]. During this period, plants experience oxidative stress, associated with excess O_2_, resulting in cellular damage. Depending on the duration of the reoxygenation process, when the water level slowly decreases, in the case of the poor adaptation of the plant to the reoxygenation process, especially at an early stage of development, cellular damage can be significant and can lead to death. To facilitate the recovery of wheat after hypoxia, we used the Qu treatment (Figure 11).

Treatment with Qu after hypoxia promotes root elongation and an increase in shoot height in the Orenburgskaya 22 variety, as well as an increase in Chls content, although there is no complete restoration to the control level. Especially, after stress, treatment with Qu effectively affects the shoots of the Orenburgskaya 22 variety. Against the background of the inactivation of redox processes in the roots, these processes are significantly activated in the shoots. Perhaps the Orenburgskaya 22 variety requires a longer period to fully recover from hypoxia. But, even based on these indicators, we can talk about the positive effect of treating the Orenburgskaya 22 variety, which is sensitive to O_2_ deficiency, with Qu.

Treatment of the Zolotaya variety with Qu during reoxygenation gives a different effect. If the morphometric parameters remain practically unchanged, then the Chls content decreases both in comparison with control shoots and with shoots under hypoxia. Moreover, almost all participants in redox processes are inactivated in both shoots and roots. Consequently, this experiment also confirms the conclusion that the Zolotaya variety is a wheat variety with a low level of redox balance; O_2_ deficiency is the norm for its development and an increase in the activity of oxidative processes leads to significant damage.

Abiotic and biotic stresses cause the degradation of intracellular components through the process of autophagy [72]. Increased ROS generation under stress conditions promotes the induction of autophagy, which attenuates oxidative stress [73], thus representing a regulatory cycle. It was previously established that autophagy is induced in wheat cells of the Orenburgskaya 22 and Zolotaya varieties under stress [74] depending on the degree of damage to the cellular structures, at which a restructuring of cellular metabolism occurs and plants acclimatize to stressful conditions, or one or another variant of PCD is triggered [74,75,76]. Two pathways of plant PCD are known: “apoptosis-like”, the markers of which are DNA breaks, the release of cytochrome c from mitochondria, and the transfer of phosphatidylserine to the outer layer of the membrane; and “vacuolar death”, characterized by the formation of large vacuoles and autophagosomes [77].

O_2_ deficiency causes an increase in nuclei with DNA breaks in seedling root cells (our article). In the variety Orenburgskaya 22, breaks were found in 11% of root cells, and, in the variety Zolotaya, in 8%. Under hypoxic conditions, the root cells of Orenburgskaya 22 were noticeably larger than those of the Zolotaya variety; cells in the G2 period and prophase predominated. Many binucleated cells were observed, which indicates a violation of cytokinesis; the cytoplasm of these cells was vacuolated. The appearance of DNA breaks in the nuclei of root cells indicates the role of hypoxia as an inducer of cell death along an apoptosis-like pathway. When exposed to hypoxia, phosphatidylserine was detected on the plasma membrane of both wheat varieties, and the cell nuclei are stained with propidium iodide, which indicates their necrotic death.

The immunodetection of cytochrome c revealed the release of cytochrome c from mitochondria into the cytoplasm of cells in both wheat varieties, but, in the Orenburgskaya variety, there were significantly more (22) death cells than in the Zolotaya variety. These indicators indicate the mitochondrial pathway of apoptosis under hypoxia in two wheat genotypes. During hypoxia, giant vacuoles are formed in the Zolotaya variety, which are one of the protective mechanisms against O_2_ deficiency.

Qu activates protective mechanisms when O_2_ availability decreases only in the case of the Orenburgskaya 22 variety. Pre-treatment with Qu helps to increase the tolerance of the Orenburgskaya 22 variety. However, under the same conditions, maximum changes in cell structure are observed in the Zolotaya variety.

Thus, the use of Qu to strengthen the protective mechanisms of plants against the effects of abiotic stress is possible only after a preliminary comprehensive study of its effect on the plant.

## 4. Materials and Methods

### 4.1. Plants

Two wheat varieties Orenburgskaya 22 (*Triticum aestivum* L.) (2n = 42) and Zolotaya (*Triticum durum* Desf.) (2n = 28) were developed by the Orenburg Research Institute of Agriculture of the Steppe Ecological Detachment (Federal Scientific Center of the Russian Academy of Sciences, Orenburg, Russia). Seedlings were grown at 24 °C under a 10 h light/14 h dark photoperiod and fluorescent lamps (5000 lx). The effect of hypoxia and treatment with 0.03% Qu on the antioxidant system of two wheat varieties was studied. In an experiment on wheat seedlings, a Qu solution was used. Plants were grown in rolls [78] according to the scheme (Figure 1). The addition of Qu on wheat before (pathway A) or after stress (pathway B) allowed us to determine the protective properties of Qu for the antioxidant system. The study was carried out on six-day-old seedlings at Stage 1: Leaf development according to the BBCH system.

### 4.2. Chlorophyll Content Analysis

Chlorophyll was extracted using 80% acetone extraction [79]. Chlorophyll levels were determined by absorbance values at 665 nm and 649 nm, which were measured on an IMPLEN nanophotometer (IMPLEN, Westlake Village, CA, USA). The content of chlorophyll a and b was determined using the formulae:chl a = [(11.63 × A665) − (2.39 × A649)] × Vml/1000 × Wmg
chl b = [(20.11 × A649) − (5.18 × A665)] × Vml/1000 × Wmg
where Vml is the total volume of the extract, and Wmg is the plant weight sample.

Data were expressed as mean ± standard deviation (SD; n = 30), and significant differences were defined as *p* < 0.05. Chlorophyll determination was carried out at least three times.

### 4.3. Fluorescence Microscopy

To visualize and determine the level of ROS in the cells of root tips (4–5 mm) of six-day-old wheat seedlings, 25–50 nM carboxy-H2DFFDA (Thermo Fisher Scientific, Waltham, MA, USA) was used for 30 min, then washed three times. The samples were analyzed under an Olympus BX51 fluorescence microscope (Olympus corporation, Tokyo, Japan), ×10 objective, wavelength 490 nm. Images were obtained using Color View digital camera (Munster, Germany). ImageJ software (version 1.54i) was used to measure fluorescence intensity.

### 4.4. Light Microscopy

For 6-day-old seedlings, sections (5 mm) of the root tip were fixed in a 4% solution of paraformaldehyde (Sigma Aldrich, Saint Louis, MO, USA) in PHEM buffer, pH = 6.9 (60 mM PIPES (Sigma Aldrich, USA), 25 mM HEPES (Sigma Aldrich, USA), 10 mM EGTA (Sigma Aldrich, USA), and 2 mM MgCl2 (Sigma Aldrich, USA) for 1.5 h at room temperature. The fixative was washed with PHEM buffer. The samples were incubated 7 min in 0.4 M mannitol containing 4% cellulase (Sigma Aldrich, USA) and 5 mM EGTA, then washed with PBS buffer; after incubation in the enzyme, the roots were transferred to coverslips and separated into individual cells. The prepared preparations were dried in the refrigerator at +4 °C for 24 h and placed in Mowiol U-44 (Hoechst, Germany) with the addition of DAPI (1 μL/1 mL) (4,6-diamidino-2-phenylindole) (Sigma Aldrich, USA). The samples were analyzed under an Olympus BX51 fluorescence microscope (Japan), ×100 objective. Images were obtained using Color View digital camera (Germany). ImageJ software was used to measure fluorescence intensity.

### 4.5. Biochemical Analysis

Antioxidant activity (AOA) was determined by blocking the oxidation process of a 0.1% aqueous solution of (R)-4-[1-hydroxy-2-(methylamino)ethyl]benzene-1,2-diol (HMAEB). Absorbance was measured at λ = 347 nm [80]. Antiradical activity (ARA) was determined by the method [81] of decrease in color of a 5 × 10^−5^ M alcohol solution of 2,2-diphenyl-1-picrylhydrazyl (DPPH). Absorbance was measured at λ = 517 nm. The concentration of peroxide in aqueous solutions of plants was determined by the decrease in color of a 0.02 M KMnO4 solution. Absorbance was measured at λ = 480 nm [82]. The content of glutathione (GSH} in mM was determined by the Elman method by the appearance of color after adding a 0.01 M alcohol solution. Absorbance was measured at λ = 412 nm [83].

### 4.6. Total RNA Isolation and Gene Expression Analysis

The analysis was carried out using a standard kit for RNA isolation “Extran RNA Synthol” (Russia). RNA was isolated from the roots and shoots of wheat grown under different conditions. cDNAs were synthesized by reverse transcription using standard methods (Syntol, Moscow, Russia). The cDNA concentration was determined spectrophotometrically using an IMPLEN nanophotometer. RT-PCR using SYBR Green I (Syntol) was carried out in a CFX 96 real-time thermal cycler (BioRad, Hercules, CA, USA). Information on the structure of *Cu/ZnSOD*, *MnSOD*, *PX*, *GPX*, *CAT,* and *GST* genes in wheat was obtained from NCBI. Primers for the genes were synthesized by Syntol. The relative level of gene expression was calculated using a calibration curve constructed with PCR products obtained with primers for the *GaPDh* gene. Each RT-PCR reaction was performed in triplicate.

### 4.7. Statistical Methods

Statistica 6.0 (version 13.3.721) and statistical software package STATAN-2009 were used for statistical calculations using Student’s *t* test (R version 4.3.1). Different letters indicate significant differences at *p* < 0.05. Values are presented as means ± standard deviations of triplicate biological copies, with significant differences at *p* < 0.05.

## 5. Conclusions

Under natural conditions, plants are often exposed to various stresses that affect their growth and development. One of the visible symptoms of abiotic stress is impaired seedling development. The greatest morphometric changes are caused by the effect of hypoxia on the Orenburgskaya 22 variety (*Triticum aestivum*). Using cytological analysis, various types of cytoplasmic changes in root cells under hypoxic conditions were identified. The formation of giant vacuoles during hypoxia in the Zolotaya variety (*Triticum durum*) is one of the protective mechanisms. Various stress factors lead to an increase in ROS and increased damage to various plant tissues. Increased ROS production under abiotic stress triggers antioxidant defense in wheat. In the Orenburgskaya 22 variety, the most active components of the antioxidant system are MnSOD and Cu/ZnSOD, and, in the Zolotaya variety, GSH. The antioxidant quercetin reduces the formation of ROS. Pre-treatment of wheat seedlings with 0.3% quercetin has a protective effect against O_2_ deficiency on seedlings of the Orenburgskaya 22 variety, but has a negative effect on seedlings of the Zolotaya variety. Treatment of seedlings of the Orenburgskaya 22 variety with quercetin after exposure to hypoxia helps restore their development. Quercetin is effective and helps protect against oxidative stress for seedlings of the Orenburgskaya 22 variety. Thus, under hypoxia, quercetin has a different effect on two wheat genotypes.

This study may have practical implications for growing wheat in different climatic conditions. However, extensive further research is required on the effect of quercetin on different wheat varieties under different stress conditions.

## Figures and Tables

**Figure 1 ijms-25-04487-f001:**
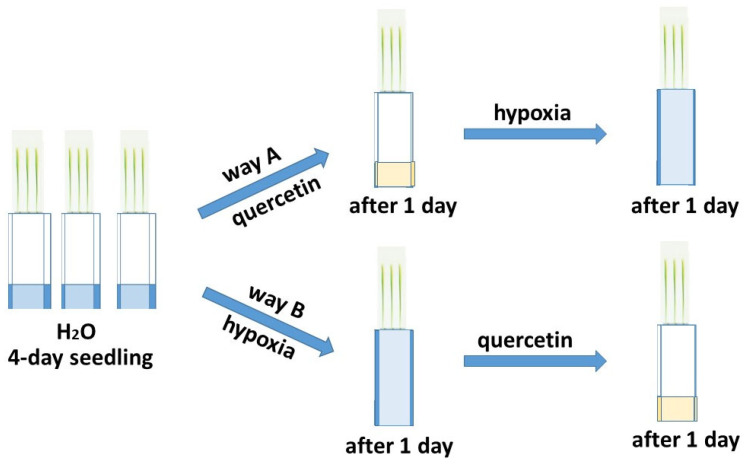
Network for treating two wheat genotypes with quercetin.

**Figure 2 ijms-25-04487-f002:**
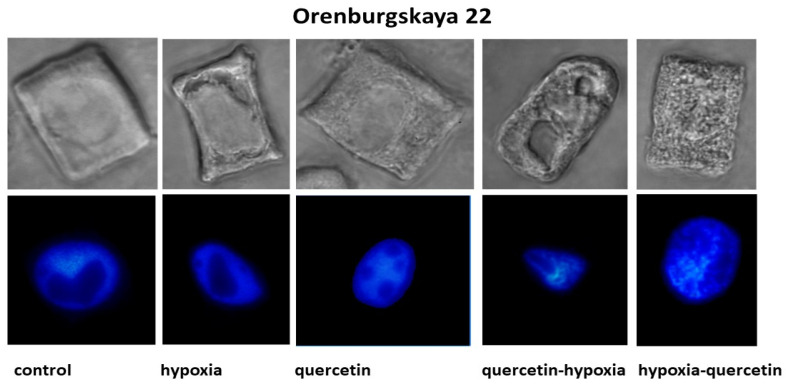
Cells with different types of altered cytoplasm in the roots of wheat varieties Orenburgskaya 22 and Zolotaya, grown under different conditions. Below are nuclei-stained with DAPI. Bar 10 μm.

**Figure 3 ijms-25-04487-f003:**
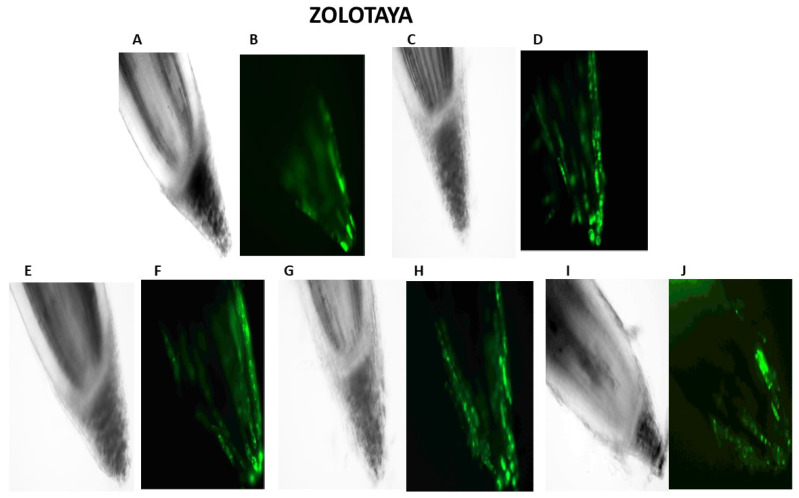
Distribution of ROS+ and ROS—cells in the zones of six-day seedling of wheat roots Orenburgskaya 22 and Zolotaya: A, B—control; C, D—hypoxia: E, F—Qu; G, H—Qu+hypoxia; and I, J—hypoxia+Qu. Bar 400 µm.

**Figure 4 ijms-25-04487-f004:**
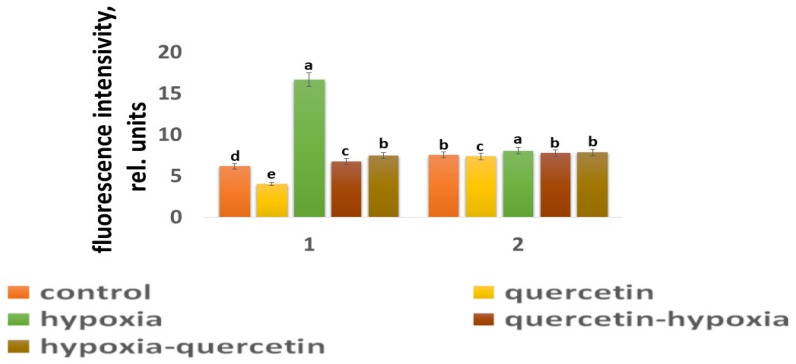
The intensity of ROS fluorescence under the influence of stress factors in wheat: 1—variety Orenburgskaya 22; and 2—variety Zolotaya. a–e-indicate significant difference were determined (*p* < 0.05).

**Figure 5 ijms-25-04487-f005:**
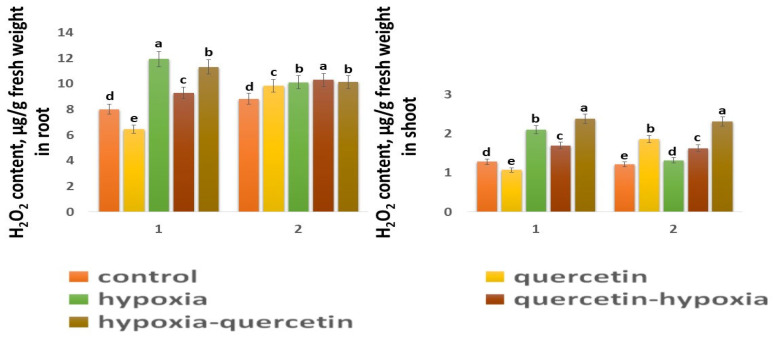
H_2_O_2_ content in roots and shoots in six-day seedlings of Orenburgskaya 22 variety (1) and Zolotaya variety (2). a–e-indicate significant difference were determined (*p* < 0.05).

**Figure 6 ijms-25-04487-f006:**
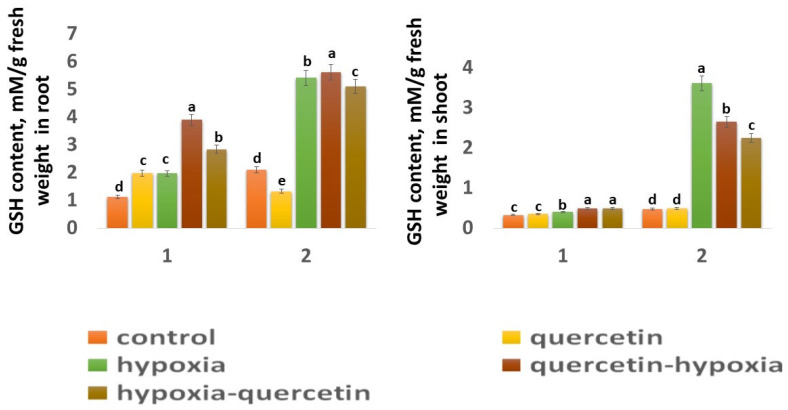
GSH content in roots and shoots in six-day seedlings of Orenburgskaya 22 variety (1) and Zolotaya variety (2). a–e-indicate significant difference were determined (*p* < 0.05).

**Figure 7 ijms-25-04487-f007:**
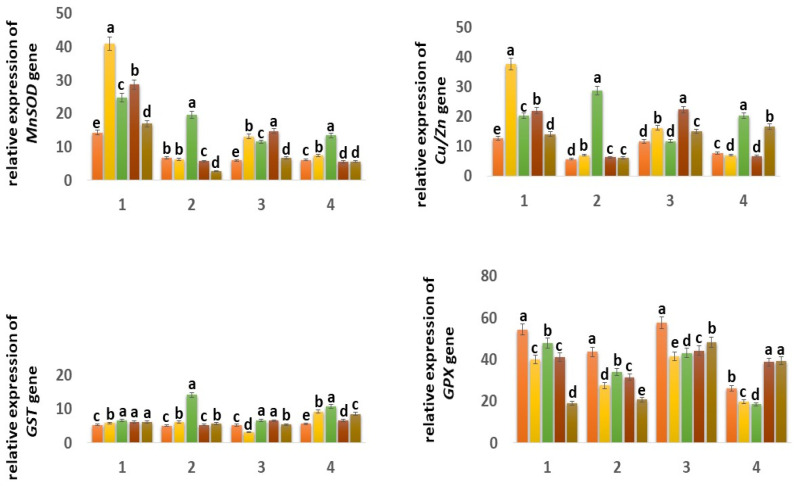
Expression of *MnSOD*, *Cu/ZnSOD*, *GST*, *GPX*, *CAT,* and *PX* genes in six-day seedlings of wheat varieties Orenburgskaya 22 (1, 3) and Zolotaya (2, 4) under the influence of various factors in (1,2)—roots, (3,4)—shoots. a–e-indicate significant difference were determined (*p* < 0.05).

**Figure 8 ijms-25-04487-f008:**
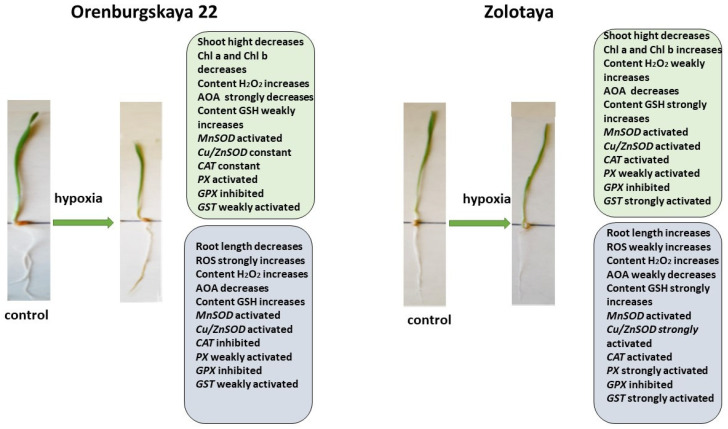
Diagram of the total changes occurring in wheat of the Orenburgskaya 22 variety and the Zolotaya variety under the influence of hypoxia relative to the control.

**Figure 9 ijms-25-04487-f009:**
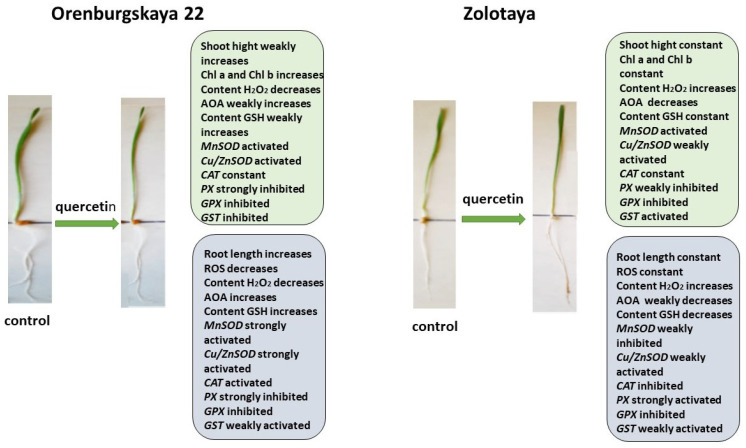
Diagram of the total changes occurring in wheat of the Orenburgskaya 22 variety and the Zolotaya variety under the influence of Qu relative to the control.

**Figure 10 ijms-25-04487-f010:**
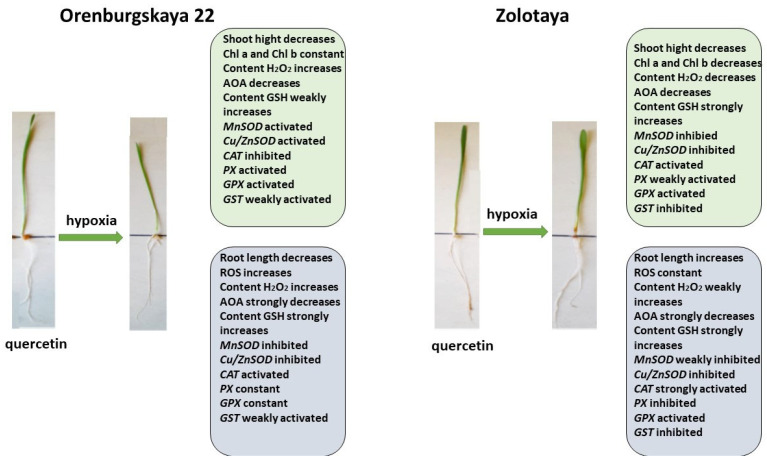
Diagram of the total changes occurring in wheat of the Orenburgskaya 22 variety and the Zolotaya variety under the influence of hypoxia relative to the Qu.

**Figure 11 ijms-25-04487-f011:**
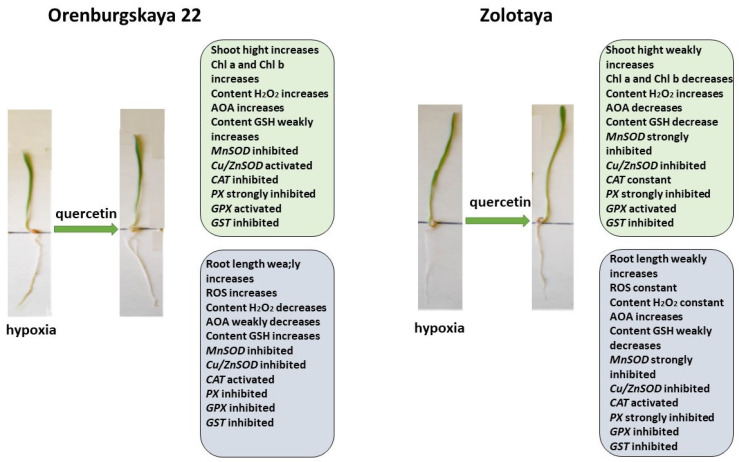
Diagram of the total changes occurring in wheat of the Orenburgskaya 22 variety and the Zolotaya variety under the influence of Qu relative to the hypoxia.

**Table 1 ijms-25-04487-t001:** Morphometric parameters of wheat varieties Orenburgskaya 22 and Zolotaya grown under different conditions.

Wheat Variety	Growth Condition	Seedling Length, cm	Shoot Hight, cm	Root Length, cm
Orenburgskaya 22	control	29.9 ± 1.44 b	17.4 ± 0.87 a	12.5 ± 0.62 b
	quercetin	30.8 ± 1.54 a	17.7 ± 0.88 a	13.1 ± 0.65 a
	hypoxia	25.0 ± 1.3 d	14.1 ± 0.7 c	10.9 ± 0.54 d
	quercetin–hypoxia	27.6 ± 1.38 c	15.3 ± 0.76 b	12.3 ± 0.61 b
	hypoxia–quercetin	26.3 ± 1.31 c	14.8 ± 0.74 c	11.5 ± 0.57 c
Zolotaya	control	27.7 ± 1.38 a	16.0 ± 0.8 a	11.7 ± 0.58 b
	quercetin	27.8 ± 1.39 a	16.2 ± 0.81 a	11.6 ± 0.58 b
	hypoxia	26.9 ± 1.34 a	13.2 ± 0.66 b	13.7 ± 0.68 a
	quercetin–hypoxia	27.0 ± 1.35 a	13.6 ± 0.68 b	13.4 ± 0.67 a
	hypoxia–quercetin	27.5 ± 1.37 a	13.6 ± 0.68 b	13.9 ± 0.69 a

Data were expressed as mean ± standard deviation (SD; n = 30) and a–d-indicate significant difference were determined (*p* < 0.05).

**Table 2 ijms-25-04487-t002:** The number of cells with altered cytoplasm in the wheat varieties Orenburgskaya 22 and the Zolotaya variety under different growing conditions per 100 cells.

Growth Condition	Orenburgskaya 22	Zolotaya
control	6 ± 2	8 ± 2
hypoxia	70 ± 8	60 ± 8
quercetin	8 ± 1	15 ± 3
quercetin–hypoxia	15 ± 2	70 ± 5
hypoxia–quercetin	25 ± 3	35 ± 3

**Table 3 ijms-25-04487-t003:** Content of Chl a and Chl b in wheat Orenburgskaya 22 and Zolotaya grown under different conditions.

Wheat Variety	Growth Condition	Chl a, mg/g	Chl b, mg/g	Ch a/Chl b
Orenburgskaya 22	control	4.13 ± 0.2 b	1.69 ± 0.08 b	2.44 ± 0.12 c
	quercetin	5.09 ± 0.25 a	1.79 ± 0.09 a	2.93 ± 0.15 a
	hypoxia	1.25 ± 0.06 d	0.46 ± 0.02 d	2.72 ± 0.13 b
	quercetin–hypoxia	5.06 ± 0.25 a	1.76 ± 0.09 a	2.87 ± 0.14 a
	hypoxia–quercetin	2.60 ± 0.13 c	1.03 ± 0.05 c	2.53 ± 0.13 c
Zolotaya	control	2.47 ± 0.12 b	0.88 ± 0.04 a	2.89 ± 0.14 a
	quercetin	2.44 ± 0.12 b	0.92 ± 0.05 a	2.66 ± 0.13 b
	hypoxia	2.70 ± 0.13 a	0.92 ± 0.05 a	2.93 ± 0.15 a
	quercetin–hypoxia	1.28 ± 0.06 d	0.48 ± 0.02 b	2.65 ± 0.13 b
	hypoxia–quercetin	1.41 ± 0.07 c	0.50 ± 0.02 b	2.83 ± 0.14 a

Data were expressed as mean ± standard deviation (SD; n = 30) and a–d-indicate significant difference were determined (*p* < 0.05).

**Table 4 ijms-25-04487-t004:** Antiradical activity (ARA) and antioxidant activity (AOA) in the roots and shoots of Orenburgskaya 22 and Zolotaya wheat varieties.

Wheat Variety	Growth Condition	Antiradical Activity, % (Method DPPH)	Antioxidant Activity, % (Method HMAEB)
Orenburgskaya 22	control	56.86 ± 2.84 b	39.04 ± 1.95 b
root	quercetin	69.55 ± 3.48 a	46.14 ± 2.31 a
	hypoxia	43.63 ± 2.18 c	25.98 ± 1.3 c
	quercetin–hypoxia	38,88 ± 1.94 d	25.12 ± 1.25 c
	hypoxia–quercetin	42.85 ± 2.14 c	21.37 ± 1.07 d
Orenburgslaya 22	control	52.08 ± 2.6 a	36.15 ± 1.81 a
shoot	quercetin	53.52 ± 2.67 a	37.37 ± 1.87 a
	hypoxia	36.74 ± 1.84 d	23.24 ± 1.16 c
	quercetin–hypoxia	43.22 ± 2.16 c	32.21 ± 1.61 b
	hypoxia–quercetin	46.44 ± 2.32 b	18.93 ± 0.95 d
Zolotaya	control	54.24 ± 2.71 a	35.86 ± 1.79 a
root	quercetin	53.4 ± 2.67 a	34.93 ± 1.75 a
	hypoxia	49.57 ± 2.48 b	31.56 ± 1.58 b
	quercetin–hypoxia	36.59 ± 1.83 d	31.78 ± 1.59 b
	hypoxia–quercetin	46.51 ± 2.32 c	24.77 ± 1.24 c
Zolotaya	control	53.42 ± 2.67 a	37.12 ± 1.85 a
shoot	quercetin	49.82 ± 2.49 b	35.94 ± 1.8 b
	hypoxia	47.05 ± 2.35 c	33.74 ± 1.69 c
	quercetin–hypoxia	42.66 ± 2,13 d	38.04 ± 1.9 a
	hypoxia–quercetin	41.09 ± 2.05 d	30.21 ± 1.51 d

Data were expressed as mean ± standard deviation (SD; n = 30) and a–d-indicate significant difference were determined (*p* < 0.05).

## Data Availability

Data are contained within the article.

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
