# Peer review of "Features of the Effect of Quercetin on Different Genotypes of Wheat under Hypoxia"

_ijms, 2024, doi:10.3390/ijms25084487_

Round 1

Reviewer 1 Report

Comments and Suggestions for Authors

he submitted manuscript is very interesting and stimulating in the context of studying the effect of biostimulants on plants growing under stress conditions. Hypoxia, which is not a standard experimental stressor, was chosen as the stress factor. For this reason, I believe that the results obtained will be beneficial and will serve for further research. The manuscript is written relatively carefully, but Russian sentences appear occasionally in the text (line 758). Please check the text. The chapters build on each other. In the introductory section, I somewhat miss the influence of phytohormones, since, for example, ethylene allows some plants in hypoxia and anoxia to form aerenchyma by dissolving the cell wall. The results are based on the description of the graphs and tables. It would be useful to adjust the quality of the graphs. The results describe the measured values in differences of multiples, but these are not shown in the tables and graphs, it would be useful to focus on using the actual measured values. In the section on oxidative stress, enzymes, etc., citations are already given in the text. These are the results. Is this necessary? It is therefore necessary to modify the text. The discussion is adequate, although descriptive in places. The methodological procedures are adequate and meet the objectives of the experiment. I would only recommend a description of the growing conditions (temperature, lighting, light regime). Furthermore, perhaps a brief description of the selected varieties. The age of the plants was given in days, perhaps it would have been more appropriate to add the developmental stage BBCH. It is necessary to unify the citation of sources, as journals are written in full title and in abbreviated form. Older literature tends to predominate. Is it necessary? 

Author Response

Thank you for your high and friendly assessment of our research. Thank you for your comments and recommendations for improving the manuscript.

Reviewer 2 Report

Comments and Suggestions for Authors

Main comments:

The authors of the manuscript addressed the interesting topic of the impact of stress caused by oxygen deficiency and the use of quercetin on the growth of seedlings of two wheat genotypes, Triticum aestivum and Triticum durum. I consider the comparative studies of selected wheat varieties performed by the authors to be valuable. They carried out the necessary tests of growth parameters, microscopic tests and the concentration of selected physiological substances used to determine the response of plants to stress. However, I have some comments regarding the way the manuscript was written, which contains incomprehensible sentences and inaccuracies, both in the description of the experiment and in the explanation of the results obtained. Overall, the style requires improvement as well as linguistic correction.

Detailed comments and suggestions:

Abstract

The abstract contains imprecise expressions, e.g.:

Line 12-13: two wheat genotypes and two cultivars: 'Orenburgskaya 22' and 'Zolotaya' (please specify whether they are cultivars or varieties. This applies to the entire manuscript (A cultivar is purposely created by humans to enhance chosen traits through artificial selection. A variety is a version of the plant species that occurs naturally through natural selection). When choosing varieties or cultivar, please be guided by, for example, the publication: Lyapunova, O.A. Intraspecific classification of durum wheat: New botanical varieties and forms. Russ J Genet Appl Res 7, 757–762 (2017). https://doi.org/10.1134/S2079059717070048.

In English one variety (singular) two or many varieties (plural). This applies to the entire content of the manuscript variety seems to be an incorrect scientific term.

Line 18-20: Should also be the full name: ROS. As well as explained other abbreviations in this sentence.

Introduction

Line 35-36 and 39-40. Information in sentences is repeated

Line 52: Should also be full name: ROS.

Results

Line 107: Figure 1. The title of the figure is not stylistically correct in English and should contain the full name quercetin instead of the abbreviation. The diagram is too general and not fully understandable to the reader, where is the control, what exactly does this description "after 1 day" mean?

Line 108: The expression: "In Table 1 shows the morphometric parameters" is not valid

Line 203: Tables 3. Should be the full name of chlorophyll

Line 434-36: Figure 7. What units are the vertical values ​​shown in?

Line 547-551: Figure 8 and 9, as well as 10 and 11. Chl(space) a and b.

Line 556-557: Figure 11. Lack of chlorophyll b.

Materials and Methods

Line 741: Wheat cultivars. Italicized: Triticum aestivum and Triticum durum

Line 744-745: The sentence is incorrect. How was oxygen deficiency stress - hypoxia prepared? How long did the experiments last? How many seeds were sown? What constituted control? How many days after sowing the seeds was quercetin applied?

Line 750-751: How many samples were taken to measure the chlorophyll content?

Line 752: "IMPLEN nanophotometer" device manufacturer

Line 756: state what "W" stands for

Line 758: No residue needed

Line 785: Determined by the method... What method?

Conclusions

Line 808-811: The first two sentences seem unnecessary in this chapter.

Comments on the Quality of English Language

The manuscript should be stylistically corrected and appropriate word order and punctuation should be used. Grammar needs to be improved.

Author Response

Thank you for your careful reading of the manuscript and valuable comments.

Reviewer 3 Report

Comments and Suggestions for Authors

The manuscript contains quite interesting research results on hypoxia in two wheat varieties, Orenburgskaya (Triticum aestivum L.) and Zolotaya (Triticum durum L.).

I ask the authors to indicate the practical use of the research results they obtained. What should be the aim of further research?

Comments

Line 12, correct Triticum aestivum L. and Triticum durum L.

Line 744, in which year was the research carried out?

Line 744, please give brief characteristics of the two wheat varieties tested

Line 804, please provide full details of the producer of the statistical software used to compile the data

References, please remove publications older than 10 years and especially from the 20th century.

Author Response

Thank you for your high and friendly assessment of our research. Thank you for your comments and recommendations for improving the manuscript

Round 2

Reviewer 2 Report

Comments and Suggestions for Authors

The authors of the manuscript improved the text and added necessary information as indicated in the review. A minor note regarding linguistic correctness. Line 205: Table 3: Content of... or Chlorophyll a nad b content.  

Comments on the Quality of English Language

Minor stylistic errors in sentence structure.

Author Response

Thank you for your comment. We agree that sometimes the construction of a sentence is not entirely correct. The next stage of working with the manuscript will be working with an editorial proofreader. We hope some errors will be corrected.